# Most Monogenic Disorders Are Caused by Mutations Altering Protein Folding Free Energy

**DOI:** 10.3390/ijms25041963

**Published:** 2024-02-06

**Authors:** Preeti Pandey, Emil Alexov

**Affiliations:** Department of Physics and Astronomy, Clemson University, Clemson, SC 29634, USA; preetip@clemson.edu

**Keywords:** monogenic disorders, protein stability, folding free energy, mutation, pathogenicity

## Abstract

Revealing the molecular effect that pathogenic missense mutations have on the corresponding protein is crucial for developing therapeutic solutions. This is especially important for monogenic diseases since, for most of them, there is no treatment available, while typically, the treatment should be provided in the early development stages. This requires fast targeted drug development at a low cost. Here, we report an updated database of monogenic disorders (MOGEDO), which includes 768 proteins and the corresponding 2559 pathogenic and 1763 benign mutations, along with the functional classification of the corresponding proteins. Using the database and various computational tools that predict folding free energy change (ΔΔG), we demonstrate that, on average, 70% of pathogenic cases result in decreased protein stability. Such a large fraction indicates that one should aim at in silico screening for small molecules stabilizing the structure of the mutant protein. We emphasize that knowledge of ΔΔG is essential because one wants to develop stabilizers that compensate for ΔΔG, but do not make protein over-stable, since over-stable protein may be dysfunctional. We demonstrate that, by using ΔΔG and predicted solvent exposure of the mutation site, one can develop a predictive method that distinguishes pathogenic from benign mutations with a success rate even better than some of the leading pathogenicity predictors. Furthermore, hydrophobic–hydrophobic mutations have stronger correlations between folding free energy change and pathogenicity compared with others. Also, mutations involving Cys, Gly, Arg, Trp, and Tyr amino acids being replaced by any other amino acid are more likely to be pathogenic. To facilitate further detection of pathogenic mutations, the wild type of amino acids in the 768 proteins mentioned above was mutated to other 19 residues (14,847,817 mutations), the ΔΔG was calculated with SAAFEC-SEQ, and 5,506,051 mutations were predicted to be pathogenic.

## 1. Introduction

The advent of next-generation sequencing has transformed the approach to investigating and diagnosing human genetic disorders. Next-generation sequencing, in conjunction with bioinformatics pipelines, has enabled the efficient analysis of a large number of DNA variants found throughout entire genomes. However, this wealth of data presents challenges when identifying individual molecular effects contributing to diseases. Accurately pinpointing one or two pathogenic mutations in Mendelian diseases is crucial amidst numerous harmless variants that naturally occur in the human genome. This requires distinguishing true positives (actual disease-causing mutations) and false positives (benign mutations). On average, an individual’s exonic genome region contains approximately 20,000 variations [1,2]. Among these, only a small fraction of non-synonymous changes in DNA sequences could be directly involved/associated with disorders. Thus, identifying and prioritizing mutations based on their pathogenic potential is of significant value for precision medicine and diagnostics.

This prompted the development of numerous in silico methods to predict the pathogenicity of amino acid mutations [3,4,5,6,7,8,9,10,11,12,13,14,15,16,17,18,19,20]. Most of these methods use amino acid features to assess the likelihood of a mutation being pathogenic. These features include evolutionary conservation, physical and chemical properties of amino acids, and some include structural information like B-factor, etc. While these methods have been successful in distinguishing between a benign and pathogenic mutation, none of the methods provide a comprehensive understanding of the functional consequence of a mutation to assist the development of therapeutic solutions. This is not crucial for complex diseases, which are caused by multiple mutations in multiple genes, as mutations can be mapped onto affected pathways or interaction networks, thus pinpointing the effect causing the disease. In the case of monogenic disorders, the pathogenicity is caused by a single mutation in a single gene, and revealing the molecular effect of the mutation is crucial for the development of treatment. Furthermore, complex diseases affect many individuals, and thus, a variety of therapeutic solutions were developed, while monogenic diseases are typically rare, and there are no therapeutic solutions for many of them. This motivated us to focus on monogenic disorders and to investigate what is the dominant molecular effect causing such disorders.

Proteins perform their function by adopting a particular 3D structure and binding mode, and any deviation from the native structural features may alter the protein’s function. This prompted another set of investigations and developments, to link pathogenic mutations with changes in protein folding and binding free energies. Although limited, attempts have been made to understand both the effect of mutations on protein stability and the potential impact on the interaction of proteins with other macromolecules. Recent research conducted an extensive analysis of mutations, considering experimental measurements of changes in protein folding and binding free energy. The study revealed a strong correlation between the probability of a mutation being pathogenic and its impact on protein folding or binding free energy (with a Pearson correlation coefficient, PCC, of up to 0.7) [21]. A similar observation was reported by other researchers; however, in their study, the benign mutations were artificially created [22]. Another investigation used computational stability predictors on a dataset consisting of 3338 pathogenic and 10,170 benign mutations from ClinVar and gnomAD v2.1. This study showed that these predictors could provide valuable insights into the potential pathogenicity of mutations, with an area under the curve value of 0.66 [23]. In a recent study, Aledo et al. developed a fitness–stability model based on the Arrhenius law to investigate the impact of mutations on protein stability. The model was validated through the extensive mutational analysis of 14,094 proteins and demonstrated that most mutations destabilize protein structure [24]. Furthermore, they observed a positive correlation between the extent of destabilization caused by amino acid substitutions and their potential for causing diseases [24]. With regard to the interaction of proteins with other biomolecules such as DNA, RNA, and other proteins, research has demonstrated that mutations can significantly hinder the ability of a protein to bind and interact with these molecules effectively. This can have implications for the development of various diseases, including cancer, cystic fibrosis, and cardiovascular disorders, among others [25,26,27,28]. The proportion of mutations stabilizing proteins is relatively low, comprising less than one-third of the available data. It is worth noting that while increased thermostability of proteins can be advantageous in preventing thermal inactivation and conformational changes at higher reaction temperatures, it should also be recognized that certain pathogenic mutations may result in protein stabilization. The H101Q mutation in the chloride intracellular channel 2 (CLIC2) protein is an example of this observation [29,30]. This variant has demonstrated a higher stability level than the wild-type protein. Functional investigations on the H101Q variant have revealed its ability to enhance, rather than hinder, ryanodine receptor (RyR) intracellular channel activity, resulting in prolonged channel openings and the potential amplification of calcium signals that depend on RyR channel functionality [29]. The R111G and A140V variants of the methyl-CpG binding protein 2 (MeCP2) are mutations that increase protein stability but significantly decrease DNA binding [31]. It is important to note that these studies did not differentiate between monogenic and polygenic disorders. However, the results of these studies clearly indicate that there is a correlation between the stability of proteins and their potential for causing diseases. Therefore, it remains an open question as to what extent these predictions can be generalized to monogenic diseases. Considering this, in our earlier work, we reported a database of monogenic disorders consisting of two datasets: Dataset 1 consists of 686 proteins and 1934 pathogenic and 1405 benign mutations (only mutations that are classified as pathogenic or benign), and Dataset 2 consists of 768 proteins and 2559 pathogenic and 1763 benign mutations (Dataset 2 includes the likely benign and likely pathogenic cases as well) [32]. These datasets allowed us to explore further the relationship between missense mutations and their impact on protein stability and pathogenicity in monogenic disorders. Our study revealed a strong association between changes in the folding free energy and the potential pathogenicity of mutations. By averaging the folding free energy values using various folding energy predictors for two datasets, we obtained an area under the curve value of 0.71 and Matthews correlation coefficient value of 0.32 [32].

The change in protein stability due to a mutation is directly related to the solvent exposure of the mutation site. It is expected that mutations occurring in the core of the protein will affect protein stability more than mutations at the protein’s surface. In a recent study, the authors explored the association between changes in solvent accessibility and the pathogenicity of human protein variations [33]. Utilizing an in silico approach, they predicted the changes in solvent accessibility (∆SASA) for a large dataset of residues undergoing variations in 12,494 human protein sequences lacking three-dimensional structures. Overall, 69,385 OMIM (Online Mendelian Inheritance in Man) related single residue variations were considered, out of which 39,436 were neutral and 29,949 were disease-related. They found that the majority of the disease-causing mutations occur primarily in the buried positions (67%) compared to the neutral ones (64.3%), which tend to occur in the exposed region [33]. We want to emphasize again that these findings were obtained without differentiating between monogenic and polygenic disorders.

Moreover, the impact of missense mutations on protein function extends beyond changes in stability alone. The consequences of missense mutations on protein function are multi-faceted and, apart from stability changes, can involve disruptions of protein–protein interactions, protein–DNA/RNA interactions, active site changes, and conformational dynamics of protein. These disruptions ultimately can lead to abnormal protein function and contribute to the development of diseases. Therefore, it is important to consider how a missense mutation affects a protein’s functionality beyond its stability. However, accurately quantifying the impact of missense mutations on stability changes is relatively straightforward, through the estimation of folding free energy change. However, assessing other properties, such as protein interaction with other macromolecules, active site changes, and alterations in conformational dynamics, becomes more challenging and complex. The complexity arises due to the incomplete characterization of all interacting partners and active site residues for each protein. Considering this, in this study, we specifically examine changes in folding free energy as a measure of pathogenicity. The goal of the current work is (a) to enrich the MOGEDO database of monogenic disorders with functional annotation; (b) to determine if the leading folding free energy change predictors can discriminate between pathogenic and benign mutations; (c) to understand whether the solvent-accessible surface area of mutation sites can be used as a measure of pathogenicity; and (d) to compare the performance of a simple machine learning-based predictor that uses physics-based quantities, i.e., folding free energy change and SASA, as features with the leading pathogenicity predictors. We also show how the association between a change in folding free energy and a mutation to become pathogenic varies among different families of the protein and the chemical nature of the mutations. This is important for drug design, to develop drugs that bind to mutants and restore wild-type folding free energy. Moreover, the insights acquired from investigating monogenic diseases can also be employed to elucidate the complex interplay between genes and environmental factors involved in the development and advancement of polygenic disorders.

## 2. Results

### 2.1. Database of Monogenic Disorders (MOGEDO)

In our earlier work, we compiled a comprehensive database of monogenic disorders, which contains information on genetic mutations associated with various monogenic diseases. The database includes information on the types of mutations, their locations within the genes, and the corresponding clinical phenotypes. Here, we provide a comprehensive overview of the database, in terms of different types of amino acid changes based on the chemical nature of the amino acid. The results are provided in Appendix A for Dataset 1 and Appendix A for Dataset 2. In both the datasets, most of the mutations have been made from Arg (Dataset 1: 553; Dataset 2: 763) to other amino acids, followed by mutations from Gly (Dataset 1: 302; Dataset 2: 385), Ala (Dataset 1: 266; Dataset 2: 340), Leu (Dataset 1: 224; Dataset 2: 277), and Ser (Dataset 1: 224; Dataset 2: 270). It is worth mentioning that the majority of the mutations in both datasets primarily consist of mutations to Arg (Dataset 1: 346; Dataset 2: 434), Ser (Dataset 1: 303; Dataset 2: 369), Val (Dataset 1: 256; Dataset 2: 310), Pro (Dataset 1: 239; Dataset 2: 287), and Leu (Dataset 1: 217; Dataset 2: 294). However, when it comes to commonly utilized datasets for training models that estimate changes in folding free energy, protein–protein binding free energy, and protein–DNA binding free energy, most of the mutations are made to alanine. This is because alanine scanning is widely recognized as one of the most popular methods for examining how mutations lead to alterations. Interestingly, mutations of Cys, Gly, Arg, Trp, and Tyr to any other amino acids are likely to result in pathogenicity (Appendix A).

Considering the properties of amino acids, we see more cases of small-to-small, followed by polar-to-polar, large-to-large, and hydrophobic-to-hydrophobic mutations (Appendix A). The analysis of amino acid changes in our database of monogenic disorders revealed certain patterns based on the chemical nature of the amino acids. For instance, in the case of pathogenic mutations, there is a higher prevalence of mutations from hydrophobic to polar amino acid and polar to hydrophobic amino acid. Interestingly, mutations from negatively charged amino acids to positively charged amino acids are more likely to be pathogenic; the vice versa is less likely to be pathogenic.

### 2.2. Change in Folding Free Energy and Pathogenicity

In our earlier work, we showed that there is a significant correlation between the changes in folding free energy and pathogenic mutations. Considering all predictors, the average folding free energy change yielded an area under the curve value of 0.71 for MOGEDO datasets. However, by combining predictions from SAAFEC-SEQ and INPS-SEQ (Impact of Non-synonymous mutations on Protein Stability) predictors to calculate the average folding free energy change, an improved AUC of 0.77 was observed. The increase in AUC could be due to the utilization of diverse features by these methods, resulting in an expanded set of input features that captures a broader range of information, contributing to change in folding free energy as a result of mutation. It should be emphasized that simply increasing the number of predictors does not necessarily result in improved performance, as there may be overlap or redundancy among the features used by each predictor, which may overfill the model and lead to overfitting. Therefore, it is essential to carefully select and combine predictors that provide complementary information for an accurate prediction of the changes in folding free energy as a result of mutation.

A few studies have shown absolute ΔΔG to be a better indicator of pathogenicity, as it treats stabilization and destabilization equally [23,34,35]. Nevertheless, no improvement in AUC was observed for any of the predictors in both datasets (Appendix A). This is not surprising, since only some mutations result in stabilization (Table 1). On the contrary, we see a significant decrease in the area under the curve when predictions from individual methods were used. A negligible effect on the AUC (Dataset 1: 0.70) was seen when the average change in folding free energy was used to calculate the AUC.

Here, we present the distribution of the change in folding free energy using various predictors for the MOGEDO database. It is important to note that a negative value for the change in folding free energy indicates a destabilizing mutation, whereas a positive value indicates a stabilizing mutation. The results clearly show that most mutations are destabilizing based on all predictors (Figure 1 and Appendix A). This observation holds for the entire dataset and when benign and pathogenic mutations are analyzed separately. There is considerable overlap between the folding free energy profiles of benign and pathogenic mutations. However, on average, pathogenic mutations tend to have a more significant negative change in folding free energy than benign mutations. This suggests that changes in folding-free energy can be used as an indicator of pathogenicity. Pathogenic mutations tend to make the corresponding protein less stable; in contrast, benign mutations have less impact on protein stability.

In the literature, a cut-off value of 1.0 kcal/mol or 2 kcal/mol has been commonly used to distinguish between mutations strongly affecting protein stability and those that do not. A similar cut-off is typically used to suggest “hot spots”, by mutating wild-type residue to Ala. Presumably, a mutation in a hot spot will dramatically affect both protein structure and function and is likely to be pathogenic. However, typically, the usage of either 1 kcal/mol or 2 kcal/mol is not justified. Our results indicate that the cut-off value differs for various predictors and is not a universally applicable threshold for all the predictors. The optimal Matthews correlation coefficient (MCC) value for distinguishing benign from pathogenic mutations using SAAFEC-SEQ was achieved with a cut-off value of 1.1 kcal/mol. For I-mutant 2.0 and INPS-SEQ, the best MCC values were obtained with cut-off values of 1.7 kcal/mol and 0.5 kcal/mol, respectively. When averaging the change in folding free energy across all predictors, the best MCC was observed at a cut-off value of 0.7 kcal/mol. Table 1 shows the count of stabilizing and destabilizing mutations in the monogenic disorder database using different predictors and cut-off values commonly found in the literature, as well as the average cut-off value (average of ΔΔG cut-off value for the three predictors), based on the optimal MCC value. It can be seen that the ratio of destabilizing to stabilizing mutations is significantly higher across all predictors and at all cut-off values.

### 2.3. Folding Free Energy Change as a Measure of Pathogenicity Based on the Chemical Nature of Amino Acid Mutations

In the earlier section, we have shown that changes in folding free energy can be used as a potential indicator of pathogenicity. In this section, we wanted to understand how the change in folding free energy as a measure of pathogenicity varies based on the chemical nature of amino acid mutations. Since different amino acid mutations can have varying effects on protein stability and the methods to estimate the change in folding free energy have been trained on a diverse set of mutations, it is plausible that the ability of folding free energy change to discriminate pathogenic from benign mutations may differ for different types of amino acid changes. To investigate this, we categorized the amino acid mutations in the monogenic disorder database into the following different groups based on their chemical nature: hydrophobic–hydrophobic, hydrophobic–polar, polar–polar, polar–hydrophobic, small–small, small–large, large–large, large–small, aliphatic–aliphatic, aliphatic-aromatic, aromatic–aromatic, aromatic–aliphatic, positive–positive, positive-negative, negative–negative, and negative–positive. The count of amino acid mutations in each category is given in Appendix A. We only considered those categories for analysis where the number of cases exceeds 100. The receiver operating characteristics (ROC) plot for the change in folding free energy as a measure of pathogenicity based on the chemical nature of amino acid mutations is shown in Figure 2 for Dataset 1 and in Appendix A for Dataset 2. It is worth noting that the ability of the folding free energy change to predict pathogenicity can vary depending on the specific chemical characteristics of amino acid mutations. Consistently, the average of folding free energy predicted from SAAFEC-SEQ and INPS-SEQ gives the best AUC for all categories, followed by INPS-SEQ alone. The ability of I-mutant 2.0 is limited in differentiating between benign and pathogenic mutations for all categories.

The best AUC was obtained for the hydrophobic–hydrophobic category, with a value of 0.84, using the average of the change in free energy value calculated using SAAFEC-SEQ and INPS-SEQ and INPS-SEQ, respectively, followed by small–large and large–large mutations for which the values are 0.83, using the average from SAAFEC-SEQ and INPS-SEQ for Dataset 1. The worst performance is obtained for the polar–hydrophobic mutation, followed by the small–small mutation. Also, the AUC obtained for these categories is poorer than the AUC of the whole of Dataset 1. A similar trend is also observed for Dataset 2, where the best AUC is obtained for the hydrophobic–hydrophobic category, followed by small–large mutation.

### 2.4. Folding Free Energy Change as a Measure of Pathogenicity Based on Functional Category

In addition to examining the chemical characteristics of amino acid mutations, it is also important to consider the functional category of the proteins when predicting their pathogenicity. To investigate how folding-free energy change as an indicator of pathogenicity varies among different functional categories, we conducted an analysis considering different types of proteins. Both datasets are dominated by the enzyme functional category, followed by the transport/translocation/cargo protein and transcriptional regulation. The underrepresented categories in the datasets include antigen–antibody, RNA binding, and membrane protein (Appendix A). It is to be noted here that we only considered those categories for further analysis where the total number of cases is at least greater than 100. The ROC plot in Figure 3 and Appendix A illustrates the performance of change in folding free energy as a predictor for pathogenicity based on the specific functional categories of proteins. The analysis revealed that the ability of change in folding free energy to predict pathogenicity can vary depending on the functional category of proteins. Similar to what has been observed in the earlier section, the average of folding free energy predicted from SAAFEC-SEQ and INPS-SEQ and INPS-SEQ alone gives the best AUC. Again, the performance of I-mutant 2.0 is low for all categories. Also, the ability of folding free energy change to discriminate pathogenic and benign mutations varies among different functional categories. SAAFEC-SEQ and I-mutant 2.0 performed best for receptor proteins with AUC values of 0.78 and 0.62, respectively, while INPS-SEQ for scaffold proteins had an AUC value of 0.80. In the case of transport/translocation/cargo protein, we see a significant decrease in the correlation between folding free energy and a mutation to be pathogenic.

### 2.5. Relative Surface Area (RSA): A Measure of Pathogenicity

Another critical factor that has been associated with the pathogenicity of single amino acid variants is the relative surface area of the mutated residue [33]. However, this property has rarely been included in the physicochemical characteristics adopted to describe the residues undergoing variations [34,36,37]. To understand the correlation between the relative surface area of mutated residues and pathogenicity, we followed the same protocol as was followed for the change in folding free energy. The analysis demonstrated that relative surface area strongly correlates with pathogenicity, with an area under the curve value of 0.78 for both Dataset 1 and Dataset 2 (Appendix A). The best MCC was obtained at a cut-off of 0.35. These findings suggest that the relative surface area of mutated residues can serve as a useful predictor for the pathogenicity of monogenic disorders, and buried residues are more frequently associated with pathogenicity compared to solvent-exposed residues. Further, we also calculated the correlation between the change in folding free energy calculated using different methods and the relative surface area of mutated residues. The two quantities showed a very weak correlation, indicating that the change in folding free energy and relative surface area could independently contribute to the pathogenicity of single amino acid variants (Appendix A).

### 2.6. Logistic Regression Model Training and Testing

The logistic regression model was trained on datasets of monogenic disorders, and the results are presented in Table 2. The best performance in distinguishing pathogenic mutations from benign mutations was achieved by training the regression model using the average change in folding free energy predicted by SAAFEC-SEQ and INPS-SEQ, along with relative surface area (RSA) as a feature. This model showed an AUC value of 0.84 and an MCC value of 0.55 for both the training set and test set of Dataset 1, followed by the model trained using only the change in folding free energy predicted by INPS-SEQ and RSA. Similar results were obtained for Dataset 2 as well. Hence, it can be concluded that utilizing changes in folding free energy alone, along with RSA, yields better performance for discriminating between pathogenic and benign mutations across both datasets.

### 2.7. Performance Comparison of Change in Folding Free Energy Method, RSA Method, and Logistic Regression Model with Other Leading Pathogenicity Predictors

The performance of the change in the folding free energy method as a predictor of pathogenicity was compared to other leading pathogenicity predictors, including the logistic regression model trained on the monogenic disorder database. We determined the optimal cut-off values for predicting folding free energy and RSA by calculating the MCC values across various cut-off values of ΔΔG and RSA. The threshold was selected based on the ΔΔG cut-off that yielded the highest MCC, to differentiate between pathogenic and benign mutations. A similar protocol was followed for the RSA method. The best accuracy was obtained for PolyPhen, using the classifier model trained on the HumVar dataset (0.86 ± 0.01), followed by the PolyPhen classifier model trained on HumDiv (average accuracy: 0.83 ± 0.01) (Table 3) for Dataset 1. The RSA method and the linear regression models trained in this work show comparable accuracy with the leading pathogenicity predictors, PhD-SNP (Predictor of human Deleterious Single Nucleotide Polymorphisms) and SIFT 4G (Sorting Tolerant From Intolerant For Genomes); however, the true positive and false negative rates for SIFT 4G are impressive compared to all methods. An improvement is seen in accuracy, true positive rate (TPR), and false negative rate (FNR) when the change in folding free energy and RSA are used together to predict the pathogenicity of a mutation. The same is true for Dataset 2.

### 2.8. Profiling Pathogenic Mutations through Folding Free Energy Change Estimated Using SAAFEC-SEQ

As we have shown in the earlier section that folding free energy change can be used as a measure of pathogenicity, we further explored SAAFEC-SEQ to profile pathogenic mutations by mutating all the sites of a protein to other 19 amino acids. This was undertaken for all the proteins in the MOGEDO database. Pathogenic mutations were identified using the cut-off value of −1.1 kcal/mol. Out of 14,847,817 mutations performed, 5,506,051 mutations are predicted to be pathogenic (the results of the predictions are available for download from http://compbio.clemson.edu/lab/downloads/, accessed on 1 October 2023). This indicates that a significant proportion of amino acid mutations have the potential to cause disease via destabilizing the corresponding protein, and therapeutic solutions should focus on the development of small molecule stabilizers [38,39].

## 3. Discussion

Understanding the molecular mechanisms of diseases is crucial for developing effective diagnostic tools and therapeutic interventions, as has been demonstrated in numerous studies [40,41,42,43,44,45,46]. Genetic disorders are frequently caused by missense mutations in specific proteins, leading to malfunctioning proteins and subsequent disease phenotypes. Since the human population consists of a plethora of genetic variations, it is essential to identify and distinguish between pathogenic and benign mutations to accurately diagnose and treat disorders. Although plenty of methods are available in the literature to predict the pathogenicity of mutations accurately, most of the methods do not provide a comprehensive understanding of the functional implications of these mutations. Since mutations can have complex functional consequences—for instance, they can affect the protein folding process and the binding of proteins to other macromolecules like other proteins, DNA, RNA, etc.—understanding the functional effect of mutation is essential for the drug discovery process [47,48,49,50,51,52,53,54]. In the current work, we aimed to assess the pathogenicity of a mutation using a thermodynamic approach, i.e., the change in folding free energy. We focused on disorders caused by a mutation in a single gene, i.e., monogenic disorders, as these often have clearer genotype–phenotype correlations than polygenic disorders.

Here, we reported the MOGEDO database, the database of monogenic disorders, which contains information about pathogenic and benign mutations in various proteins involved in monogenic disorders, and analyzed the functional consequences of mutations by assessing the impact of the mutation on protein folding. In addition, we provided a comprehensive overview of the monogenic disorder database with reference to amino acid mutations, the chemical nature of the mutation site, how the change in folding free energy as a measure of pathogenicity varies for different classes of proteins, and how it compares with the existing leading pathogenicity predictors (Align GVGD, PhD-SNP, PolyPhen, and SIFT), including the RSA method and logistic regression models, built using the change in folding free energy and RSA as features.

A notable difference observed in the monogenic disorder database was that mutations involving Cys, Gly, Arg, Trp, and Tyr amino acids being replaced by any other amino acid are more likely to be pathogenic. This could be due to the essential roles that these amino acids play in protein structure and function. For instance, Cys is an essential catalytic residue and is also involved in the formation of disulfide bridges, which are essential for stabilizing the structure of the protein; Gly is often found in structural regions, and its substitution can disrupt protein conformation; and Arg and Tyr are frequently found in protein binding sites, and mutations in these residues can affect protein–protein interactions. Trp is often involved in the hydrophobic core formation and protein stability. Concerning the chemical nature of amino acid mutations, mutations from negatively charged amino acids to positively charged amino acids are more likely to be pathogenic; however, vice versa are less likely to be pathogenic. Further, hydrophobic–polar, polar–hydrophobic, small–large, and large–small mutations are also more likely to be pathogenic. This analysis was prompted by the observation that mutations involving a drastic change in the chemical nature of amino acids can have severe consequences on protein structure and function. To further illustrate the linkage between pathogenicity and structural features, we selected a case for which there is a 3D structure available. ThusC163S mutation is a pathogenic mutation frequently observed in N(4)-(beta-N-acetylglucosaminyl)-L-asparaginase (AGA, PDB Id: 1APY, 1APZ), which leads to aspartylglucosaminuria (AGU), an autosomal recessive inherited disorder of glycoprotein degradation. This mutation abolishes autocatalytic cleavage and enzyme activity. C163 may be important for folding of the glycosylasparaginase protein through its involvement in a disulfide bridge. The disruption of this particular disulfide bond could lead to the misfolding of the protein, potentially causing it to become stuck in the endoplasmic reticulum due to a conformational change or an interaction between the newly freed sulfhydryl and a component within the endoplasmic reticulum.

It is important to note that the impact of amino acid substitutions on protein structure and function can vary depending on several factors. These factors include the specific protein being studied, the location of the amino acid residue within the protein structure, and the type of mutation that occurs. Therefore, it is crucial to consider these factors when studying the pathogenicity of a mutation and assessing the linkage to protein destabilization. Therefore, we evaluated the correlation between the change in folding free energy and pathogenicity across different classes of amino acid mutations, based on their chemical properties, and also among different classes of proteins. Our analysis revealed that the correlation is not consistent for all classes or across all predictors of ΔΔG. The strongest correlation was found for hydrophobic–hydrophobic mutations, using the average of folding free energy predicted from SAAFEC-SEQ and INPS-SEQ, followed by aliphatic–aliphatic mutations and hydrophobic–polar mutations. On the other hand, a weak correlation was observed for polar–hydrophobic and small–small mutations, which was worse than the performance on the overall dataset. Hydrophobic–hydrophobic mutations have a strong correlation with changes in folding free energy. These mutations preserve the hydrophobic character of the residue environment, which is critical for core stability and the three-dimensional structure of proteins. Therefore, these mutations likely maintain the hydrophobic packing without being affected by specific structural details, leading to a clear and strong correlation with changes in folding free energy. However, when mutations involve a change from a hydrophobic to a polar amino acid, the effects on protein folding energy can be more complex. Polar–hydrophobic mutations could disrupt existing polar interactions such as hydrogen bonds, again leading to destabilization. The difference in correlation obtained for the two categories, i.e., hydrophobic–polar and polar–hydrophobic mutations, might be due to the specific roles that polar residues play in a protein’s structure, which can include interactions with water or other polar molecules, like ions and hydrogen bonds, that stabilize the protein’s secondary and tertiary structures. Changes in these functions can have diverse effects on the stability of a protein, thus causing the different correlations observed. The unique environment of each protein and the location of the mutation within the protein’s three-dimensional structure further modulate the effect on protein stability, making some hydrophobic–polar or polar–hydrophobic mutations more disruptive than others. While these general trends are commonly observed, there may be multiple factors contributing to the exact details and reasons for variations in correlation, which depend on individual proteins and their specific mutation contexts.

When considering the functional category of proteins, it was found that receptor proteins exhibited slightly stronger correlations between changes in folding free energy and pathogenic mutations using all the predictors. However, for transport/translocation/cargo proteins, there was a decrease in the performance of change in folding free energy as a predictor of pathogenicity. Because these proteins primarily work by binding to other proteins or macromolecules, mutations are most likely to impact their binding affinities rather than their overall folding stability. While both receptor proteins and transport/translocation/cargo proteins involve interactions with other molecules, the nature of these interactions differs in both cases. Receptor proteins often function through direct binding to ligands or other molecules, and therefore, alterations in their structures can directly influence their ability to bind, leading to clear functional consequences and pathogenic effects. On the other hand, transport proteins and translocation/cargo proteins are involved in the movement of molecules or ions across membranes, and their function relies on specific binding and recognition of substrates and movement across cellular compartments. This multifaceted nature of their function might make it more difficult to correlate the effects of mutations, as there are other factors at play beyond binding.

Besides folding free energy change, we also explored RSA as a measure of pathogenicity and observed that it shows a strong correlation with pathogenicity with an AUC value of 0.78. This suggests that a change in the relative solvent accessibility of a protein residue can serve as an effective indicator of its pathogenicity. Further, the logistic regression model developed, which utilizes folding free energy change and RSA as features, demonstrated comparable accuracy to leading pathogenicity predictors such as PhD-SNP and SIFT 4G and is indeed better than some pathogenicity predictors (Align GVGD). While PolyPhen achieved the highest accuracy overall, SIFT 4G showed impressive rates for correctly identifying positive cases and minimizing false negatives compared to all other methods. It should be mentioned that pathogenicity can result from changes in other properties of the protein affecting its function. Approaches incorporating evolutionary data along with features from amino acid sequences and/or structures indirectly account for these aspects, leading to better predictions of pathogenicity. However, the method developed in this work focuses on folding free energy change and RSA, which is based on first principle physics, capturing one specific aspect: pathogenicity resulting from the change in folding stability and solvent accessibility. Nonetheless, the strength of this method lies in providing insights into the underlying cause of mutations and their effects on protein structure, which is valuable for understanding functional consequences and designing targeted therapies.

In summary, our research provides evidence to support the use of physics-based measurements, such as changes in folding free energy and solvent accessibility, as reliable indicators of pathogenicity. These metrics can effectively assist in identifying variants that may have pathological implications. Further, this approach provides a more comprehensive understanding of the effect of mutation on protein function, as it also reveals the underlying cause for the mutation to be pathogenic and the magnitude of its impact on protein stability. This information is essential from the drug discovery perspective as it can aid in developing targeted therapies. For example, when designing a small molecule to target a specific mutated protein, understanding the folding stability of the protein can be crucial. This information helps determine if the mutation is likely to stabilize or destabilize the protein structure, which can guide the design process. Furthermore, the magnitude with which the mutation affects the protein’s stability can inform decisions on the optimal drug design strategy. Suppose one aims to develop a small molecule for a destabilized mutant protein and restore its wild-type folding free energy. In that case, it is important that any additional stabilization provided by the small molecule must align with or closely match the change in folding free energy caused by the mutation. Over-stabilizing proteins may lead to dysfunction and disease manifestation, as demonstrated in the literature [29,30].

It is interesting to note that approximately 70% of the pathogenic mutations result in decreased protein stability, implying that the loss of structural integrity is a common mechanism underlying disease development. There are opposing views in the literature regarding the major cause of pathogenicity, with some studies emphasizing the disruption of protein–protein interactions as the primary cause [55]. Perhaps the outcome depends on the database being used to carry out the investigation. In our case, the MOGEDO database was created without any bias toward the corresponding proteins (being or not strong folders or being strong or not binders), but rather focusing on a balanced set (equal number of benign and pathogenic mutations) in genes associated with monogenic disorders.

We also examined the possible disease-causing mutations for all proteins in the MOGEDO database. This was carried out by systematically mutating each site to 19 different amino acids and utilizing SAAFEC-SEQ to calculate the resulting changes in folding free energy. Out of 14,847,816 mutations performed, approximately 37% were found to be potentially pathogenic based on our calculations of changes in folding-free energy. This list of potential disease-causing mutations can help prioritize further investigations and experimental validations to confirm the pathogenicity of these variants.

## 4. Materials and Methods

### 4.1. Database of Monogenic Disorders (MOGEDO)

To conduct our study on the correlation between changes in protein folding free energy and the likelihood of a mutation being pathogenic, we used the database of monogenic disorders [32]. For the development of the database, the list of genes was obtained from OMIM [56], where only genes associated with monogenic diseases were selected, and cancer-related diseases were excluded. A total of 3108 genes were chosen to collect missense mutations. The mutations for each gene were retrieved from ClinVar [57]. Only genes containing mutations annotated as benign, pathogenic, likely benign/benign, or likely pathogenic/pathogenic were considered for the dataset. If the percentage of the benign mutations and the pathogenic mutations of a gene was larger than 10%, the mutations of that gene were saved in Dataset 1. If the proportion of benign and pathogenic mutations in a particular gene was greater than 10%, the mutations associated with that gene were included in Dataset 1. The same criterion was applied to create Dataset 2, but including mutations that were annotated as likely benign and likely pathogenic. Subsequently, around 14,000 variations were selected for further analysis. The dataset was additionally filtered based on population frequency data obtained from the Ensembl genome database [58]. Variants classified as benign with a population frequency below 0.01 and variants classified as pathogenic with a population frequency above 0.01 were excluded. Additionally, if there were discrepancies in clinical significance between the data sources, those particular mutations were excluded from the dataset. As a final result, Dataset 1 consisted of 686 proteins and 1934 pathogenic and 1405 benign mutations, and Dataset 2 consisted of 768 proteins and 2559 pathogenic and 1763 benign mutations. The MOGEDO database includes information like the RefSeq accession ID of corresponding proteins, allele ID, OMIM gene name, ENSEMBLE gene ID, OMIM phenotypes, and experimental conditions. The database was further enriched to provide the functional classification of the proteins listed. The proteins were classified into 16 distinct groups. These groups included enzyme, transport/translocation/cargo protein, transcription regulation, structural support, scaffold protein, receptor protein, signaling protein, regulatory protein, DNA binding, motor protein, secretory proteins, adhesion protein, chaperons, membrane protein, RNA binding, and antigen–antibody complexes. Any remaining unclassified proteins were grouped under miscellaneous. The MOGEDO database is available to download from http://compbio.clemson.edu/lab/downloads/ (accessed on 1 October 2023).

For the calculation of folding free energy change as a result of mutation, the protein sequences corresponding to the genes were obtained from the NCBI RefSeq sequence database. It is important to note that not all mutations in a given gene were present in the same isoform. To ensure consistency, we selected one specific isoform for all mutations; however, if a mutation was observed only in a particular isoform, then that specific isoform was used for calculating the change in folding free energy caused by the mutation.

### 4.2. Folding Free Energy Calculation

To estimate the impact of mutations on folding free energy, we utilized three sequence-based methods: SAAFEC-SEQ [59], I-mutant 2.0 [60], and INPS-seq [61]. We only focused on sequence-based methods because there are a sufficient number of cases in the monogenic disorder dataset where protein structure remains unknown. Additionally, we selected these methods because of their widespread use, accessibility, and user-friendly nature. It should be noted that the change in folding free energy was calculated differently from that of the change in binding free energy. Therefore, a negative value of ΔΔG (ΔG_wt_ − ΔG_mutant_) indicated de-stabilization, while a positive value indicated stabilization.

Below, we briefly describe the methods used to estimate change in folding free energy.

SAAFEC-SEQ [59]: SAAFEC-SEQ is a machine learning method based on gradient-boosting decision trees, which incorporates physicochemical properties, sequence features, and evolutionary information to estimate the impact of amino acid mutations on folding free energy. This method requires amino acid sequences as input for prediction.

I-mutant 2.0 [60]: I-mutant 2.0 is implemented as both sequence and structure base methods that utilize support vector machine algorithms to predict changes in folding free energy resulting from mutations.

INPS-MD [61]: The INPS-MD method has been developed and implemented as both a sequence and structure-based method. It employs machine learning techniques based on support vector regression.

### 4.3. Solvent Accessible Surface Area Calculation

The relative solvent accessible surface area (RSA) of each residue undergoing mutation was calculated using NetSurfP-2.0 [62], a sequence-based method for the computation of SASA. The approach utilizes a model comprising convolutional and long short-term memory networks that have been trained using resolved three-dimensional protein structures.

### 4.4. Regression Model Development

For predicting whether a mutation is pathogenic or benign, we trained a regression model using the sklearn library in Python (version 1.4.0). The change in folding free energy and the RSA values were used as input features. In order to construct a reliable and robust regression model, an equal number of benign and pathogenic mutations were taken, and the dataset was split into a training set (80%) and a test set (20%). The training set was used to train the model and evaluate its performance, while the test set was used to validate its predictions. To build a robust model, fivefold cross-validation was performed 100 times using different c values, which, in logistic regression, is a critical hyper-parameter that controls the regularization strength of the algorithm to prevent overfitting. The final model was trained on the complete training and validation set using the c value, for which the best average score (AUC score) was obtained. Multiple regression models were trained independently for both datasets, using folding free energy change predictions from each predictor and RSA values.

### 4.5. Methods for Predicting the Pathogenicity of Amino Acid Mutations

Altogether, we employed four different methods to predict the pathogenicity of amino acid mutations for the monogenic disorder database. In the following section, we will provide a brief overview of these methods. While numerous prediction methods are available, the chosen approaches were selected for their popularity, user-friendly interfaces, and ease of installation.

Align GVGD: Align GVGD [17,18] is a widely used method for predicting the pathogenicity of amino acid mutations. It combines sequence alignment and Grantham variation and Grantham deviation scores to classify mutations as deleterious or benign.

PhD-SNP: PhD-SNP [63] is another commonly used method for predicting the pathogenicity of amino acid mutations. It combines information from protein sequence, evolutionary conservation, and the predicted secondary structure to score the potential pathogenicity of mutations.

PolyPhen: PolyPhren [5] is a computational algorithm that predicts the potential impact of amino acid substitutions on protein function. It utilizes sequence alignments and structural features of 3D proteins to assess the potential impact of amino acid substitutions.

SIFT 4G: SIFT 4G [4] is a widely used tool for predicting the pathogenicity of amino acid mutations in protein sequences. It uses a combination of sequence conservation and protein structure information to predict the impact of amino acid mutations on protein function.

### 4.6. Receiver Operating Characteristics (ROC)

To quantify the association between folding free energy and the classification of a mutation as benign or pathogenic, we assessed the change in folding free energy caused by each mutation. We assigned four categories to entries in the monogenic database: true positive, for pathogenic mutations correctly classified as such; true negative, for benign mutations accurately classified as benign; false positive, for benign mutations wrongly labeled as pathogenic; and false negative, for pathogenic mutations mistakenly identified as benign. The ROC curve analysis was conducted by adjusting cutoff values, with an area under the curve (AUC) calculated accordingly. The change in folding free energy, computed using SAAFEC-SEQ [59], I-mutant 2.0 [60], and INPS-seq [61], was used to calculate ROC and AUC values independently. Additionally, we determined the averages of SAAFEQ-SEQ and I-mutant 2.0 and of SAAFEC-SEQ and INPS-seq to obtain further metrics on ROC curves and AUCs.

### 4.7. Sampling and Assessment of Predictions

To compare how the change in folding free energy method as a predictor of pathogenicity performs compared to other commonly used methods, we performed performance evaluation using the measures: true positive rate (TPR), false positive rate (FPR), false negative rate (FNR), and accuracy. Since most of the leading pathogenicity predictors give the result mainly in the form of classifications (benign or pathogenic), we used a threshold ΔΔG to classify the mutations as pathogenic or benign for all the change in folding free energy predictors and RSA. The ΔΔG cut-off, at which the maximum Mathews correlation coefficient was observed in our dataset, was selected as the threshold for each predictor.

For performance evaluation, we considered an equal number of cases of benign and pathogenic mutations in the sample such that they are equally represented (sample size = N (benign) + N (pathogenic), where “N” is the minimum of 50% of the benign or pathogenic mutations), calculated TPR, FPR, FNR, and accuracy for each sample, and repeated the analysis 100 times to obtain the average and standard deviation. Then, we reported the average TPR, FPR, FNR, and accuracy.

## Figures and Tables

**Figure 1 ijms-25-01963-f001:**
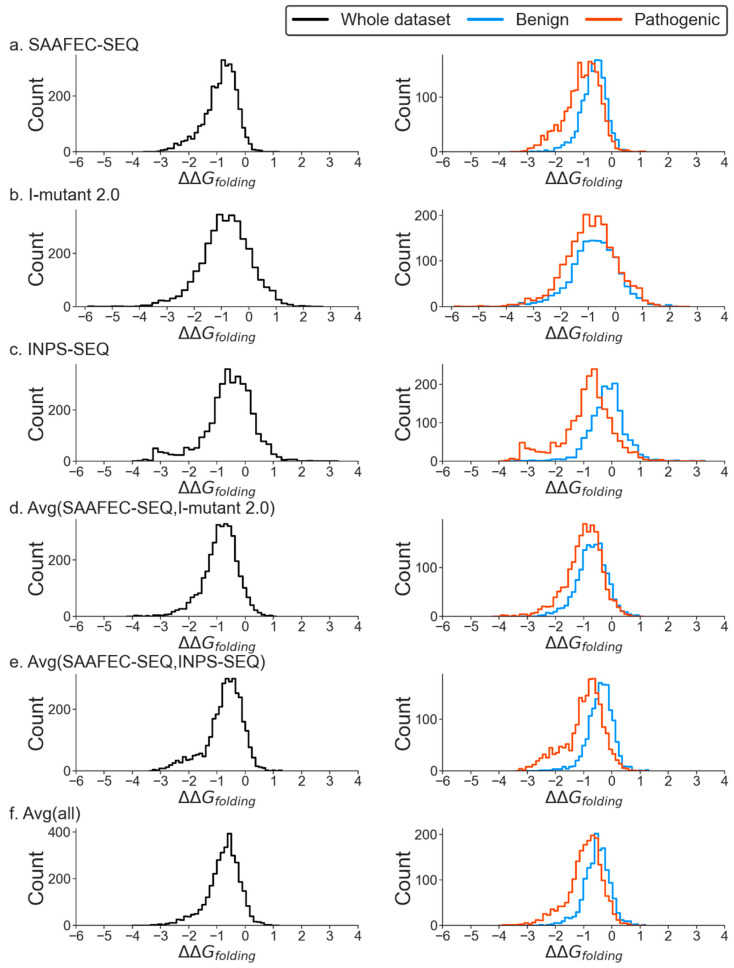
Distribution of change in folding free energy using different predictors for monogenic disorder dataset 1 (no likely cases).

**Figure 2 ijms-25-01963-f002:**
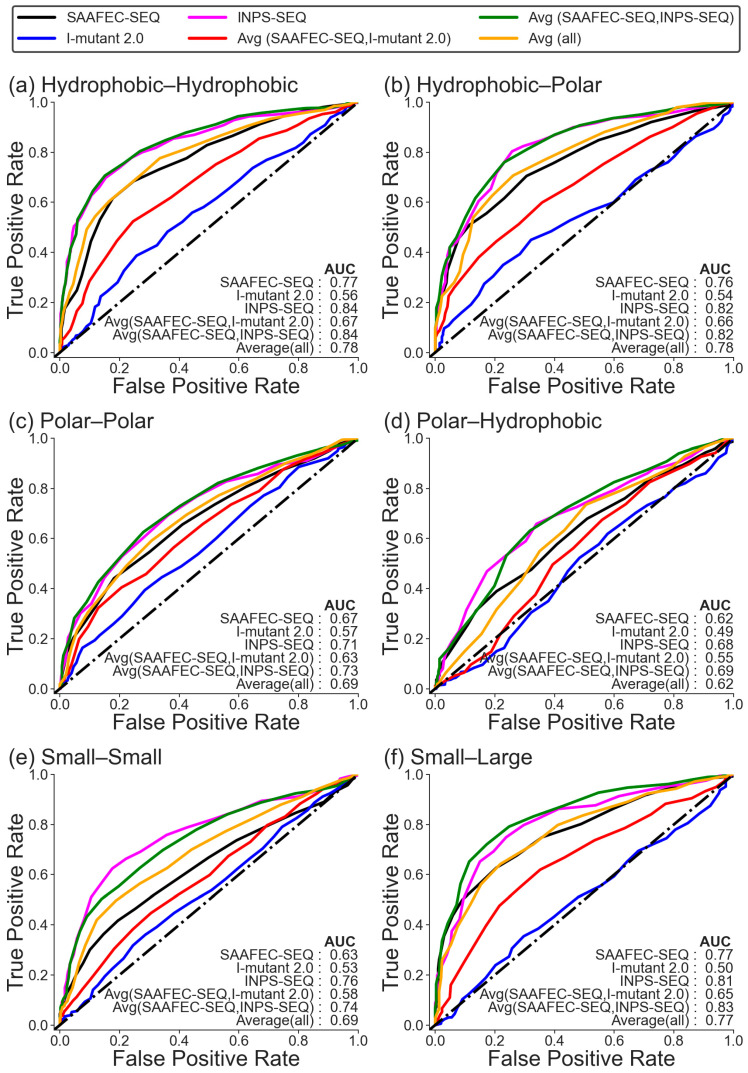
ROC curve for monogenic disorder dataset 1 (no likely cases) for different categories of amino acid mutations. According to their chemical properties, the amino acids are categorized as follows: Ala, Cys, Gly, Ile, Leu, Met, Phe, Pro, Trp, and Val are categorized as hydrophobic; Asp, Glu, Lys, Arg, His, Asn, Gln, Ser, Thr, and Tyr are categorized as polar; His, Phe, Trp, and Tyr are aromatic; Ala, ILe, Lys, Leu, Met, Pro, and Val are aliphatic; His, Lys, and Arg are positive; Asp and Glu are negative; Ala, Cys, Gly, Ser, Asn, Asp, Pro, Thr, and Val are small; and Arg, Gln, Glu, His, ILe, Leu, Lys, Met, Phe, Trp, and Tyr are large amino acids.

**Figure 3 ijms-25-01963-f003:**
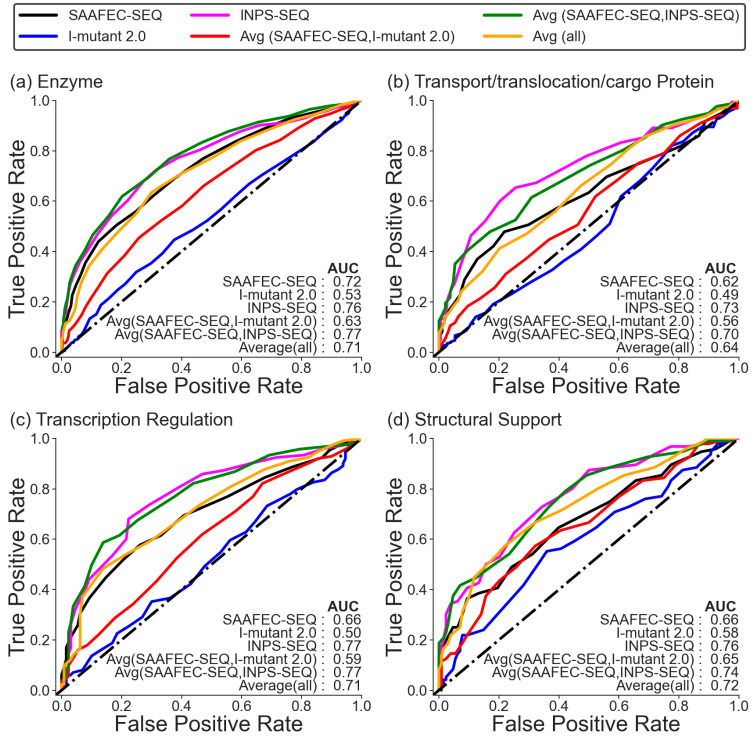
ROC curve for Monogenic Disorder Dataset 1 (no likely cases) for different functional categories.

**Table 1 ijms-25-01963-t001:** Total number of stabilizing and destabilizing mutations in the monogenic disorder database.

Methods	Dataset 1					
	Cut-off > 2 kcal/mol		Cut-off > 1 kcal/mol		Cut-off > 1.1 kcal/mol	
	No. of stabilizing mutations	No. of destabilizing mutations	No. of stabilizing mutations	No. of destabilizing mutations	No. of stabilizing mutations	No. of destabilizing mutations
SAAFEC-SEQ	0	199	1	1222	1	1021
I-mutant 2.0	3	309	44	1374	34	1223
INPS-SEQ	5	252	52	826	40	674
Avg (SAAFEC-SEQ, I-mutant 2.0)	0	141	0	1211	0	1000
Avg (SAAFEC-SEQ, INPS-SEQ)	0	214	2	883	2	734
Avg (all)	0	153	1	943	1	766
	Dataset 2					
	Cut-off > 2 kcal/mol		Cut-off > 1 kcal/mol		Cut-off > 1.1 kcal/mol	
	No. of stabilizing mutations	No. of destabilizing mutations	No. of stabilizing mutations	No. of destabilizing mutations	No. of stabilizing mutations	No. of destabilizing mutations
SAAFEC-SEQ	0	251	1	1564	1	1312
I-mutant 2.0	4	417	66	1756	48	1564
INPS-SEQ	6	327	67	1101	50	885
Avg (SAAFEC-SEQ, I-mutant 2.0)	0	190	1	1554	0	1285
Avg (SAAFEC-SEQ, INPS-SEQ)	0	280	2	1160	2	968
Avg (all)	0	207	1	1227	1	1002

**Table 2 ijms-25-01963-t002:** Performance of the logistic regression model on training and test sets for Dataset 1 and Dataset 2.

Se. No	Regression Model	Dataset 1			
		Training set		Test set	
		AUC	MCC	AUC	MCC
1.	*P* _SAAFEC-SEQ,RSA_	0.82	0.52	0.81	0.52
2.	*P* _I-mutant 2.0,RSA_	0.80	0.52	0.76	0.52
3.	*P* _INPS-SEQ,RSA_	0.84	0.54	0.85	0.54
4.	*P* _Avg (SAAFEC-SEQ, I-mutant 2.0),RSA_	0.80	0.49	0.81	0.49
5.	*P* _Avg (SAAFEC-SEQ, INPS-SEQ),RSA_	0.84	0.55	0.85	0.55
6.	*P* _Avg (All),RSA_	0.82	0.51	0.81	0.51
		Dataset 2			
		Training set		Test set	
		AUC	MCC	AUC	MCC
1.	*P* _SAAFEC-SEQ,RSA_	0.81	0.51	0.81	0.51
2.	*P* _I-mutant 2.0,RSA_	0.80	0.51	0.78	0.51
3.	*P* _INPS-SEQ,RSA_	0.84	0.53	0.83	0.53
4.	*P* _Avg (SAAFEC-SEQ, I-mutant 2.0),RSA_	0.79	0.47	0.82	0.47
5.	*P* _Avg (SAAFEC-SEQ, INPS-SEQ),RSA_	0.84	0.54	0.83	0.54
6.	*P* _Avg (All),RSA_	0.82	0.51	0.80	0.51

**Table 3 ijms-25-01963-t003:** Performance comparison of change folding free energy method, RSA method, and logistic regression model with leading pathogenicity predictors on MOGEDO.

Se. No.	Methods	Dataset 1				
		Cut-off	Avg TPR	Avg FPR	Avg FNR	Avg Accuracy
1.	SAAFEC-SEQ	−1.1	0.44 ± 0.02	0.14 ± 0.01	0.56 ± 0.02	0.65 ± 0.01
2.	I-mutant 2.0	−1.7	0.18 ± 0.01	0.12 ± 0.01	0.82 ± 0.01	0.53 ± 0.01
3.	INPS-SEQ	−0.5	0.68 ± 0.01	0.25 ± 0.01	0.32 ± 0.01	0.72 ± 0.01
4.	Avg (SAAFEC-SEQ, I-mutant 2.0)	−1.0	0.45 ± 0.02	0.26 ± 0.01	0.55 ± 0.02	0.59 ± 0.01
5.	Avg (SAAFEC-SEQ, INPS-SEQ)	−0.6	0.69 ± 0.01	0.28 ± 0.01	0.31 ± 0.01	0.71 ± 0.01
6.	Avg (all)	−0.7	0.64 ± 0.01	0.32 ± 0.01	0.36 ± 0.01	0.66 ± 0.01
7.	RSA	0.35	0.77 ± 0.01	0.27 ± 0.01	0.23 ± 0.01	0.75 ± 0.01
8.	*P* _SAAFEC-SEQ,RSA_	--	0.74 ± 0.01	0.24 ± 0.01	0.26 ± 0.01	0.75 ± 0.01
9.	*P* _I-mutant 2.0,RSA_	--	0.75 ± 0.01	0.25 ± 0.01	0.25 ± 0.01	0.75 ± 0.01
10.	*P* _INPS-SEQ,RSA_	--	0.76 ± 0.01	0.23 ± 0.01	0.24 ± 0.01	0.77 ± 0.01
11.	*P* _Avg (SAAFEC-SEQ, I-mutant 2.0),RSA_	--	0.75 ± 0.01	0.26 ± 0.01	0.25 ± 0.01	0.74 ± 0.01
12.	*P* _Avg (SAAFEC-SEQ, INPS-SEQ),RSA_	--	0.75 ± 0.01	0.21 ± 0.01	0.25 ± 0.01	0.77 ± 0.01
13.	*P* _Avg (All),RSA_	--	0.76 ± 0.01	0.25 ± 0.01	0.24 ± 0.01	0.75 ± 0.01
14.	PhD-SNP	--	0.68 ± 0.01	0.22 ± 0.01	0.32 ± 0.01	0.73 ± 0.01
15.	PolyPhen (HumDiv)	--	0.93 ± 0.01	0.27 ± 0.01	0.07 ± 0.01	0.83 ± 0.01
16.	PolyPhen (HumVar)	--	0.89 ± 0.01	0.17 ± 0.01	0.11 ± 0.01	0.86 ± 0.01
17.	Align GVGD	--	0.65 ± 0.02	0.36 ± 0.01	0.35 ± 0.02	0.65 ± 0.01
18.	SIFT 4G	0.05	0.98 ± 0.00	0.49 ± 0.01	0.02 ± 0.0	0.75 ± 0.01
		Dataset 2				
		Cut-off	Avg TPR	Avg FPR	Avg FNR	Avg Accuracy
1.	SAAFEC-SEQ	−1.0	0.42 ± 0.01	0.14 ± 0.01	0.58 ± 0.01	0.64 ± 0.01
2.	I-mutant 2.0	−1.3	0.31 ± 0.01	0.24 ± 0.01	0.69 ± 0.01	0.54 ± 0.01
3.	INPS-SEQ	−0.4	0.74 ± 0.01	0.31 ± 0.01	0.26 ± 0.01	0.71 ± 0.01
4.	Avg (SAAFEC-SEQ, I-mutant 2.0)	−1.0	0.44 ± 0.01	0.26 ± 0.01	0.56 ± 0.01	0.59 ± 0.01
5.	Avg (SAAFEC-SEQ, INPS-SEQ)	−0.6	0.70 ± 0.01	0.28 ± 0.01	0.30 ± 0.01	0.71 ± 0.01
6.	Avg (all)	−0.7	0.64 ± 0.01	0.32 ± 0.01	0.36 ± 0.01	0.66 ± 0.01
7.	RSA	0.35	0.77 ± 0.01	0.27 ± 0.01	0.23 ± 0.01	0.75 ± 0.01
8.	*P* _SAAFEC-SEQ,RSA_	--	0.75 ± 0.01	0.24 ± 0.01	0.25 ± 0.01	0.75 ± 0.01
9.	*P* _I-mutant 2.0,RSA_	--	0.76 ± 0.01	0.26 ± 0.01	0.24 ± 0.01	0.75 ± 0.01
10.	*P* _INPS-SEQ,RSA_	--	0.76 ± 0.01	0.22 ± 0.01	0.24 ± 0.01	0.77 ± 0.01
11.	*P* _Avg (SAAFEC-SEQ, I-mutant 2.0),RSA_	--	0.75 ± 0.01	0.26 ± 0.01	0.25 ± 0.01	0.75 ± 0.01
12.	*P* _Avg (SAAFEC-SEQ, INPS-SEQ),RSA_	--	0.76 ± 0.01	0.22 ± 0.01	0.24 ± 0.01	0.77 ± 0.01
13.	*P* _Avg (All),RSA_	--	0.76 ± 0.01	0.25 ± 0.01	0.24 ± 0.01	0.76 ± 0.01
14.	PhD-SNP	--	0.69 ± 0.01	0.21 ± 0.01	0.31 ± 0.01	0.74 ± 0.01
15.	PolyPhen (HumDiv)	--	0.93 ± 0.01	0.29 ± 0.01	0.07 ± 0.01	0.82 ± 0.01
16.	PolyPhen (HumVar)	--	0.89 ± 0.01	0.18 ± 0.01	0.11 ± 0.01	0.86 ± 0.01
17.	Align GVGD	--	0.66 ± 0.01	0.36 ± 0.01	0.34 ± 0.01	0.65 ± 0.01
18.	SIFT 4G	0.05	0.98 ± 0.00	0.50 ± 0.01	0.02 ± 0.00	0.74 ± 0.01

## Data Availability

All datasets are freely available for download from http://compbio/clemson.edu/ (accessed on 1 October 2023).

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
