# Peer review of "Most Monogenic Disorders Are Caused by Mutations Altering Protein Folding Free Energy"

_ijms, 2024, doi:10.3390/ijms25041963_

Round 1

Reviewer 1 Report

Comments and Suggestions for Authors

In this manuscript titled” Most monogenic disorders are caused by mutations altering protein folding free energy”, the authors mentioned about their in-silico methodology to detect the pathogenic mutants that cause the most monogenic disorders. In this manuscript, they have used the database of monogenic disorders (MOGEDO), which contains 768 proteins. In their further analysis, they have analyzed 2559 pathogenic and 1763 benign mutations, along with their functional classification of these corresponding proteins. The authors have shown a strong correlation between the decrease in the protein stability over mutation with the pathogenesis of that protein. With this huge data set, they tried to prove that the decrease in stability is associated with the pathogenicity of this corresponding mutation. They also showed that buried hydrophobic-hydrophobic interaction contributes a lot to the pathogenicity of this mutation. The author carefully observed that any mutation in the Arg, Gly, Cys, Trp, and Tyr residues to any residue resulting pathogenicity. The manuscript is well written but there are a few suggestions for the authors:

1) it is a minor thing. In most cases, the negative value of del G indicates the stability in the protein, but here authors used negative delG to indicate the instability in the protein. It would be misleading for the naïve reader. Probably, authors can think of changing the nature of the values.

2) The selection of the cut-off value is not clear. A more detailed explanation will be helpful for the readers.

3) From the observation, it is clear from Table 1 that the combination of two methods is much better than the solo one. Among them, SAAFEC-SEQ, I-mutant 2.0 combination stood better than others in the area under the curve analysis. Authors can explain it a little bit more, not with the numbers but the philosophical difference.

4) At the same time, all averages gave the best result in the AUC analysis. How do we restrict the number of parameters/ or rather number of methods to estimate it? This point is also missing.

5) The weakness or limitation of this method was not mentioned clearly. Authors may make it clear by stating their effort or pushing the limit of their methods.

6) In Figures 2 and 3, the AUC value is higher or close to 1 for SAAFEC-SEQ, INPS-SEQ method but it was stated that SAAFEC-SEQ, I-mutant 2.0 is better. Maybe I misunderstood, it is advisable to the authors to make it clear.

Author Response

We thank the reviewers for their comments and suggestions which were taken into account and appropriate charges were made. Below we address reviewers’ questions point by point.

Reviewer: In this manuscript titled” Most monogenic disorders are caused by mutations altering protein folding free energy”, the authors mentioned about their in-silico methodology to detect the pathogenic mutants that cause the most monogenic disorders. In this manuscript, they have used the database of monogenic disorders (MOGEDO), which contains 768 proteins. In their further analysis, they have analyzed 2559 pathogenic and 1763 benign mutations, along with their functional classification of these corresponding proteins. The authors have shown a strong correlation between the decrease in the protein stability over mutation with the pathogenesis of that protein. With this huge data set, they tried to prove that the decrease in stability is associated with the pathogenicity of this corresponding mutation. They also showed that buried hydrophobic-hydrophobic interaction contributes a lot to the pathogenicity of this mutation. The author carefully observed that any mutation in the Arg, Gly, Cys, Trp, and Tyr residues to any residue resulting pathogenicity. The manuscript is well written but there are a few suggestions for the authors:

  • it is a minor thing. In most cases, the negative value of del G indicates the stability in the protein, but here authors used negative delG to indicate the instability in the protein. It would be misleading for the naïve reader. Probably, authors can think of changing the nature of the values.

Ans: We followed the nomenclature established in experimental papers and the corresponding databases. Indeed, historically the effect of a mutation on protein stability and protein binding is reported differently. Thus, the change in folding free energy caused by a mutation is calculated as ΔΔGfolding = ΔGwt – ΔGmutant, and therefore, a positive value of ΔΔG indicates a mutation that makes the protein stable, while a negative value is representative of destabilization. In contrast, the change in binding free energy is calculated as ΔΔGbinding = ΔGmutant – ΔGwt. We added a sentence to explain this on Page 17.

  • The selection of the cut-off value is not clear. A more detailed explanation will be helpful for the readers.

Ans: In the revision, we justify the selection of the cut-off value based on the Matthew correlation coefficient (MCC) with max value. The description has been added on Page 12.

From the observation, it is clear from Table 1 that the combination of two methods is much better than the solo one. Among them, SAAFEC-SEQ, I-mutant 2.0 combination stood better than others in the area under the curve analysis. Authors can explain it a little bit more, not with the numbers but the philosophical difference.

Ans: Thank you for your suggestion. The reason why the combination of the two methods works better is that they use different features, and thus, the combination expands on the input features. New text is added on Page 4.

  • At the same time, all averages gave the best result in the AUC analysis. How do we restrict the number of parameters/ or rather number of methods to estimate it? This point is also missing.

Ans: We do not want to overfill the model(s), and by averaging more and more models, we essentially increase the number of features. It should be emphasized that simply increasing the number of predictors does not necessarily result in improved performance, as there may be overlap or redundancy among the features used by each predictor, which may overfill the model and lead to overfitting. Therefore, the idea is to carefully select and combine predictors that provide complementary information for accurate prediction of the changes in folding free energy as a result of mutation. We have added new text on page 4 to describe this.

  • The weakness or limitation of this method was not mentioned clearly. Authors may make it clear by stating their effort or pushing the limit of their methods.

Ans: The method is based on DDG and RSA and thus deals with first principle parameters. We acknowledge that pathogenicity may result from alteration of other properties of the macromolecules that affect its functionality. This is the limitation of the model. In contrast, the methods utilizing evolutionary information indirectly capture all deleterious effects but do not provide an explanation of why a mutation may be pathogenic. New text is added on Page 15 and 16.

  • In Figures 2 and 3, the AUC value is higher or close to 1 for SAAFEC-SEQ, INPS-SEQ method but it was stated that SAAFEC-SEQ, I-mutant 2.0 is better. Maybe I misunderstood, it is advisable to the authors to make it clear.

Ans: In the revised manuscript, we have clearly mentioned on Page 7 and Page 10 that the average of folding free energy predicted from SAAFEC-SEQ & INPS-SEQ and INPS-SEQ alone gives the best AUC, and the performance of I-mutant 2.0 is low for all categories.

Reviewer 2 Report

Comments and Suggestions for Authors

In the manuscript titled “Most monogenic disorders are caused by mutations altering 2 protein folding free energy”, Pandey and Alexov reported an updated database of monogenic disorders (MOGEDO) by examining the thermodynamic stability of point mutations causing monogenic diseases using the database and various computational tools predicting changes in folding free energy, solvent accessible area, etc. This is an extension of their previous work (reported in the journal, Curr Opin. Struc. Biol (2023)) with a total of 768 proteins having 2559 pathogenic and 1763 benign mutations. The analysis consists of two datasets: (1) based on only mutations that are classified as pathogenic or benign and (2) based on mutations that are likely pathogenic and likely benign cases. Their analysis based on sequences demonstrated a strong association of the potential pathogenicity due to mutations with the changes in folding free energy. The authors reported decreased protein stability in 70% of cases involving pathogenic mutations. Also, it was reported that among different families, there is a varying degree of association between the change in free energy and the chemical nature of the pathogenic mutations. Mutations Cys, Gly, Arg, Trp, and Tyr were identified to be more pathogenic when replaced by any other amino acid. The study was conducted reasonably, and the write-up is well organized.

1)    For clarity, the authors must define how (in what sense) the area under the curve of the average folding free energy can be used as a predictor.

2)    In the discussion (lines 391-405), the authors describe various consequences of mutations in the residues Cys, Gly, Arg, Trp, and Tyr. It might be useful to provide a table giving some examples for such claims.

3)    Most analysis and sequence based. Would it be possible to provide some examples to verify from some available structures (if any) to ascertain some of the claims from sequence related analysis.

Author Response

We thank the reviewers for their comments and suggestions which were taken into account and appropriate charges were made. Below we address reviewers’ questions point by point.

Reviewer: 

In the manuscript titled “Most monogenic disorders are caused by mutations altering 2 protein folding free energy”, Pandey and Alexov reported an updated database of monogenic disorders (MOGEDO) by examining the thermodynamic stability of point mutations causing monogenic diseases using the database and various computational tools predicting changes in folding free energy, solvent accessible area, etc. This is an extension of their previous work (reported in the journal, Curr Opin. Struc. Biol (2023)) with a total of 768 proteins having 2559 pathogenic and 1763 benign mutations. The analysis consists of two datasets: (1) based on only mutations that are classified as pathogenic or benign and (2) based on mutations that are likely pathogenic and likely benign cases. Their analysis based on sequences demonstrated a strong association of the potential pathogenicity due to mutations with the changes in folding free energy. The authors reported decreased protein stability in 70% of cases involving pathogenic mutations. Also, it was reported that among different families, there is a varying degree of association between the change in free energy and the chemical nature of the pathogenic mutations. Mutations Cys, Gly, Arg, Trp, and Tyr were identified to be more pathogenic when replaced by any other amino acid. The study was conducted reasonably, and the write-up is well organized.

1)    For clarity, the authors must define how (in what sense) the area under the curve of the average folding free energy can be used as a predictor.

Ans: The area under the curve is used as an indicator for plausible correlation between folding free energy change (DDG) and pathogenicity and indeed such a correlation is observed. Furthermore, we compute the Matthews correlation coefficient (MCC) at different cut-off of the folding free energy change and the maximum is used to suggest the optimal cut-off of DDG(cut-off) such that mutations predicted to cause DDG larger than DDG(Cu-off) are expected to be pathogenic. Appropriate text is added on p. 12.

2)    In the discussion (lines 391-405), the authors describe various consequences of mutations in the residues Cys, Gly, Arg, Trp, and Tyr. It might be useful to provide a table giving some examples for such claims.

Ans: This is done in Fig. S1b and Fig. S2e in supplementary material, where ewe show that the number of pathogenic mutations involving replacement of Cys, Gly, Arg, Trp and Tyr is the largest compared with mutations involving other types of residues. It is indicated on p. 4 in the revision.

3)    Most analysis and sequence based. Would it be possible to provide some examples to verify from some available structures (if any) to ascertain some of the claims from sequence related analysis.

Ans: As requested, an example is added in the revision, showing that mutation of Cys residue breaks disulfide bridge and thus destabilize the protein and this is the reason for pathogenicity. Text is added on p. 15.

Reviewer 3 Report

Comments and Suggestions for Authors

In this manuscript the authors have used a huge database of proteins involved in monogenic disorders and have provided insights into the various ways that pathogenic mutations can be predicted. The different properties that have been used are, (deltadelta)G arising from single mutations wherein the amino acids were classified according to their chemical nature (hydrophobic, polar) and  size (large or small) and relative surface area (RSA). While the former deals with the change in stability due to mutations, the latter is concerned with the change in relative solvent accessibility that arises on amino acid substitutions. Both gave very good correlations with the pathogenicity prediction due to mutations. ROC plots and AUCs have also been carried out for monogenic sequences as per protein functional categories. Some of the amino acids were observed to be more pathogenic, this being related to the essential roles that these play in the structure and/or function of the protein. Profiling of pathogenic mutations by mutating one position by the other 19 amino acids was a very interesting way of providing important and necessary information. The latter is a systematic way of investigating the degree of pathogenicity of mutations. This is a very important study that correlates pathogenicity and its prediction with a couple of very important protein stability parameters.

Some of the comments below need to be worked on before the manuscript can be published.

It would be interesting to see how a correlation plot for the RSA against (deltadelta)G would look like based on the database used. I hope that same can be incorporated in the manuscript.

Figure2: The ROC plots are a bit confusing to me. Why should the hydrophobic-hydrophobic have the strongest correlation – what does this mean. Rather to me, the hydrophobic-polar or polar- hydrophobic should have had the strongest correlation. Also, why did the hydrophobic-polar and polar- hydrophobic plots have different correlation ranks? Should they have not been quite similar. These aspects need to be discussed.

Figure 3 needs to be discussed with respect to the findings in the ‘discussion’ section. For example why do receptor proteins score better than the transport proteins as both involve binding partners?

While I understand that the authors have already had previous manuscripts where they have used ROC plots and AUCs, however I think that that would help a lot of the significance of these be mentioned in the present text somewhere, such that the readers, who are not necessarily familiar with these terms, may understand readily what these imply.

Author Response

We thank the reviewers for their comments and suggestions which were taken into account and appropriate charges were made. Below we address reviewers’ questions point by point.

Reviewer: 

In this manuscript the authors have used a huge database of proteins involved in monogenic disorders and have provided insights into the various ways that pathogenic mutations can be predicted. The different properties that have been used are, (deltadelta)G arising from single mutations wherein the amino acids were classified according to their chemical nature (hydrophobic, polar) and  size (large or small) and relative surface area (RSA). While the former deals with the change in stability due to mutations, the latter is concerned with the change in relative solvent accessibility that arises on amino acid substitutions. Both gave very good correlations with the pathogenicity prediction due to mutations. ROC plots and AUCs have also been carried out for monogenic sequences as per protein functional categories. Some of the amino acids were observed to be more pathogenic, this being related to the essential roles that these play in the structure and/or function of the protein. Profiling of pathogenic mutations by mutating one position by the other 19 amino acids was a very interesting way of providing important and necessary information. The latter is a systematic way of investigating the degree of pathogenicity of mutations. This is a very important study that correlates pathogenicity and its prediction with a couple of very important protein stability parameters.

Some of the comments below need to be worked on before the manuscript can be published.

It would be interesting to see how a correlation plot for the RSA against (deltadelta)G would look like based on the database used. I hope that same can be incorporated in the manuscript.

Ans: Thank you for the suggestion. We have added the plots showing the correlation between ΔΔG and RSA in the supplementary file (Figure S7a, & S7b, Page 16,17). The two quantities show a very weak correlation. We have added text on Page to describe this.

Figure2: The ROC plots are a bit confusing to me. Why should the hydrophobic-hydrophobic have the strongest correlation – what does this mean. Rather to me, the hydrophobic-polar or polar- hydrophobic should have had the strongest correlation. Also, why did the hydrophobic-polar and polar- hydrophobic plots have different correlation ranks? Should they have not been quite similar. These aspects need to be discussed.

Ans: We agree with the reviewer. One would expect that hydrophobic-polar and polar-hydrophobic will have similar correlations; however, this isn’t the case. The difference in correlation obtained for the two categories, i.e., hydrophobic-polar and polar-hydrophobic mutations, might be due to the specific roles that polar residues play in a protein's structure, which can hydrogen bonds that stabilize the protein's secondary and tertiary structures among the others. Changes of such details could diversely effects on the stability of a protein, thus the different correlations observed. A detailed discussion has been added on Page 15.

Figure 3 needs to be discussed with respect to the findings in the ‘discussion’ section. For example why do receptor proteins score better than the transport proteins as both involve binding partners?

Ans: While both receptor proteins and transport/translocation/cargo proteins involve interactions with other molecules, the nature of these interactions differs in both cases. Receptor proteins often function through direct binding to ligands or other molecules, and therefore, alterations in their structures can directly influence their ability to bind, leading to clear functional consequences and pathogenic effects. On the other hand, transport proteins and translocation/cargo proteins are involved in the movement of molecules or ions across membranes, and their function relies on specific binding and recognition of substrates and movement across cellular compartments. This multifaceted nature of their function might make it more difficult to correlate the effects of mutations as there are other factors at play beyond binding. This has been discussed in the revised manuscript in the discussion section on Page 15.

While I understand that the authors have already had previous manuscripts where they have used ROC plots and AUCs, however I think that that would help a lot of the significance of these be mentioned in the present text somewhere, such that the readers, who are not necessarily familiar with these terms, may understand readily what these imply.

Ans: The description of the ROC and AUC is provided in section 4.6, Page 18.